# Juvenile Leaves or Adult Leaves: Determinants for Vegetative Phase Change in Flowering Plants

**DOI:** 10.3390/ijms21249753

**Published:** 2020-12-21

**Authors:** Darren Manuela, Mingli Xu

**Affiliations:** Department of Biological Sciences, University of South Carolina, Columbia, SC 29208, USA; manuelad@email.sc.edu

**Keywords:** miR156, SPL, abaxial trichome, juvenile phase, adult phase

## Abstract

Vegetative leaves in *Arabidopsis* are classified as either juvenile leaves or adult leaves based on their specific traits, such as leaf shape and the presence of abaxial trichomes. The timing of the juvenile-to-adult phase transition during vegetative development, called the vegetative phase change, is a critical decision for plants, as this transition is associated with crop yield, stress responses, and immune responses. Juvenile leaves are characterized by high levels of miR156/157, and adult leaves are characterized by high levels of miR156/157 targets, SQUAMOSA PROMOTER BINDING PROTEIN-LIKE (SPL) transcription factors. The discovery of this miR156/157-SPL module provided a critical tool for elucidating the complex regulation of the juvenile-to-adult phase transition in plants. In this review, we discuss how the traits of juvenile leaves and adult leaves are determined by the miR156/157-SPL module and how different factors, including embryonic regulators, sugar, meristem regulators, hormones, and epigenetic proteins are involved in controlling the juvenile-to-adult phase transition, focusing on recent insights into vegetative phase change. We also highlight outstanding questions in the field that need further investigation. Understanding how vegetative phase change is regulated would provide a basis for manipulating agricultural traits under various conditions.

## 1. Introduction

Plants have two major stages of development, vegetative and reproductive, with distinct lateral structures, leaves and flowers, produced at each stage [1,2,3]. The vegetative phase itself is further divided into the juvenile phase and adult phase, since many “heteroblastic” traits of leaves, such as morphology, initiation rate, length/width ratio, size, leaf base angle, gibberellic acid sensitivity, vascular system complexity, and trichome growth vary as the shoot matures [1,2,3]. For example, maize juvenile leaves lack trichomes and have abundant epicuticular wax, giving off a dull blue-green color, while adult leaves have reduced epicuticular wax and trichomes [4]. As for the model plant *Arabidopsis thaliana*, juvenile leaves have smooth margins and small rounded blades, while adult leaves are more elongated, have serrated margins, and produce trichomes on the abaxial side that juvenile leaves lack [5]. The presence of these traits at particular stages of development is essential for proper plant function. 

Vegetative phase changes have also been described in other species such as *Acacia confusa*, *Acacia colei*, *Eucalyptus globulus*, *Hedera helix*, *Quercus acutissima*, *Populus*, *Oryza sativa*, and *Nicotiana tabacum* [6,7,8,9]. The juvenile-to-adult phase change, referred to as the vegetative phase change, is unidirectional and associated with certain physiological responses [2,10,11]. For example, the age-dependent production of swollen thorns by members of the genus *Vachellia* is correlated with the expression of vegetative phase change markers [9]. These swollen thorns provide food for ants while the ants protect the plant from pathogens and herbivores. *Arabidopsis* ectopically expressing juvenile phase markers produces more lateral roots, is more tolerant to salinity and drought stress, and is more resistant to herbivores, while rice ectopically expressing adult phase markers possesses an ideal plant architecture that improves grain yield (more grains per panicle) and immunity to fungus *Magnaporthe oryzae* [11,12,13,14,15,16,17]. The variety of these developmental functions and physiological responses highlight the importance of understanding the mechanisms that control vegetative phase change and the potential to harness this knowledge to enhance agriculture. How vegetative phase change markers affect various plant stress responses has been reviewed recently [18,19]. Here, we focus on how vegetative phase change itself is regulated.

## 2. What Are the Master Regulators for Vegetative Phase Change and How Were They Identified? 

Genetic studies on vegetative phase change were first carried out in maize. These studies identified several mutants, *teopod1* (*Tp1*), *teopod2* (*Tp2*), *teopod3* (*Tp3*), and *corngrass1* (*Cg1*), with a prolonged juvenile phase (i.e., more juvenile leaves), but the molecular basis for these phenotypes remained unknown for a long time [20]. Studies in *Arabidopsis* revealed that mutants with disruptions in microRNA (miRNA) biogenesis, such as *hasty* (*hst*), *squint* (*sqn*), *hyponastic leaves1* (*hyl1*), and *argonaute7* (*ago7*), were accelerated in vegetative phase change [21,22,23,24] (Figure 1). *HST*, an ortholog of importin β-like nucleocytoplasmic transport receptor exportin 5 in mammals, is involved in the transport of miRNAs from the nucleus to the cytoplasm [22,25]. Cytoplasmic accumulation of many miRNAs, including miR156, is reduced in *hst* mutants as compared with wild type (WT), leading to reduced miR156 activity [22,24,25]. SQN, a cyclophilin 40 protein that interacts with Hsp90 in *Arabidopsis*, promotes the activity of AGO1, a key component of the RNA-induced silencing complex (RISC) [21,26,27,28]. *ZIPPY* (*ZIP*) encodes AGO7, a member of the AGO family that is part of the RISC complex [23,26]. ZIP/AGO7 preferably binds to miR390, which targets *AUXIN RESPONSE FACTOR3* (*ARF3*, also termed *ETTIN*, *ETT*) and *ARF4* [23,29,30]. All of these mutants functioning in miRNA biogenesis exhibited an accelerated vegetative phase change, suggesting that an miRNA might be a master regulator for vegetative phase change (Figure 1). miR156 expression changes during vegetative development with high expression levels in juvenile leaves and low expression levels in adult leaves. Overexpression of eight miR156 coding fragments in *Arabidopsis* produced many more juvenile leaves than WT [31]. Furthermore, maize *Cg1* was cloned and shown to encode *zma-miR156b/c* [32]. In short day conditions, WT plants produced about seven juvenile leaves (16%–17% of the vegetative leaves), while plants overexpressing *miR156a* produced about 64–74 juvenile leaves (85%–88% of the vegetative leaves). In contrast, plants overexpressing a non-cleavable miR156-target mimic (*MIM156*), which reduces the activity of miR156, did not produce any juvenile leaves after germination [10]. These data strongly suggest that miR156 acts as a key regulator for vegetative phase change with decreased expression of miR156 associated with the transition from the juvenile to the adult phase. 

miR156 has a sequence complementary to a group of *SQUAMOSA PROMOTER BINDING PROTEIN-LIKE* (*SPL*) transcription factors and it negatively regulates *SPL*s transcriptionally or post transcriptoinally [31,32,33,34]. Constitutive over-expression of miR156 in maize [32], rice [7], tomato [35], poplar [6], and tobacco [8] results in the delay of the juvenile-to-adult phase transition. Mutations that render *SPL2*, *SPL9*, *SPL10*, *SPL11*, *SPL13* and *SPL15* insensitive to miR156, thereby resulting in their over-expression, have the opposite effect [10,11,36]. This suggests that vegetative phase change in higher plants is primarily controlled by the miR156/SPL module. In *Arabidopsis*, miR156 is encoded by eight loci, *MIR156A-MIR156H*. In addition, a related miRNA, miR157 is encoded by four loci, *MIR157A*-*MIR157D*. Mature miR157 has an extra C or U at the 5′ end compared to mature miR156, but it also has sequence complementary to the same group of SPL transcription factors [31,32,33,34]. Though the overall abundance of miR157 is higher than that of miR156, genetic analysis showed that miR156 plays a more important role than miR157 in the timing of vegetative phase change. This may be explained by more miR156 being loaded onto AGO1 than miR157 [33].

Several other regulators of miRNA biogenesis with roles in vegetative phase change have been identified but so far their connection with miR156 is not clear. The miRNA biogenesis mutant *suo* was identified in a vegetative phase change mutant screen after the discovery of miR156 as the master regulator for juvenile leaf identity. SUO encodes a protein that has a bromo-adjacent homology (BAH) domain at the N-terminus and two GW (glycine and tryptophan) repeats at the C-terminus [37]. Like other miRNA biogenesis mutants, *suo* mutants exhibited an accelerated vegetative phase change. SUO is present in the nucleus and in processing bodies (P-bodies). The levels of some mature miRNAs are either unaffected or elevated in *suo* mutants, and the protein levels of these miRNA targets are unaffected, reduced, or elevated, making the mechanism of how SUO functions in miRNA biogenesis ambiguous [37] (Figure 1). *ALTERED MERISTEM PROGRAM1* (*AMP1*) encodes a putative carboxypeptidase and mutation in *AMP1* resulted in faster leaf initiation rate, suggesting late vegetative phase change [38]. AMP1 was reported to mediate translational repression of some miRNAs [39], but it does not seem to regulate the activity of miR156 in this way, since the protein levels of neither miR156 sensitive nor miR156 resistant SPL9 change in the *amp1* mutant [40]. SUPPRESSOR OF GENE SILENCING3 (SGS3) and RNA-DEPENDENT POLYMERASE6 (RDR6) are involved in trans-acting small interfering RNA (ta-siRNA) biogenesis, and mutations in these genes cause phenotypes similar to *zip* (*ago7*) [41,42]. The abundance of some small RNAs, including miR156, was reduced in *sgs3* and *rdr6*, but how ta-siRNA biogenesis regulators might affect miR156 is not clear [41]. A recent study showed that ribosomal RNA-derived siRNAs (risiRNAs) compete with miRNAs in loading to AGO1 [43]. However, this cannot explain the reduced activities of miR156 in *sgs3* and *rdr6*, which was predicted to have more miR156 loaded onto AGO1. Together, the mechanisms of how SUO, SGS3 and RDR6 are involved in vegetative phase change remain unclear.

Ten of the 16 *SPL* genes in *Arabidopsis* have a sequence complementary to miR156/157, however, they respond to changes in miR156/157 differently. miR156/157 blocks the expression of all 10 targeted *SPL*s in the first two rosette leaves since levels of miR156/157 are very high in these leaves [33]. As miR156/157 levels decline in subsequent leaves, transcripts of *SPL3/SPL9/SPL15* increase quickly. miR156 represses SPL9 both transcriptionally and translationally, while miR156 represses SPL13 mainly at the translational level [40]. Of these SPLs, SPL9, SPL13, and SPL15, were found to strongly promote vegetative phase change, while SPL2/SPL10/SPL11 also contribute [11]. 

Many proteins are involved in miRNA biogenesis. Here we summarize those that function in miR156 biogenesis, as their loss-of-function mutants have been shown to have defects in vegetative phase change. In the nucleus, *pri-miR156* is transcribed by RNA polymerase II and is processed to a stem-loop structure (pre-miR156) by the activities of DICER-LIKE1 (DCL1), SERRATE (SE), and HYPONASTIC LEAVES 1 (HYL1). DCL1, SE, and HYL1 continue to process pre-miR156 to miR156-miR156* double strands. The hydroxyl groups at the 3′ ends are methylated by HUA ENHANCER1 (HEN1). The methylated miR156-miR156* are exported from the nucleus to the cytoplasm by HST, and the miR156 guide strand is loaded onto AGO1 or AGO7 to assemble the RNA-induced silencing complex (RISC) stabilized by SQN. The miR156-RISC negatively regulates a group of SPL genes who have sequences complemtary to miR156. SUO is present in processing bodies and SGS3 and RDR6 are involved in ta-siRNA biogenesis, but it is not clear how they interfere with the activities of miR156-RISC. Dashed lines indicate indirect interaction. Arrows indicate positive regulation. Black blocks indicate methylation of the 3′ end of miR156. 

## 3. How are Traits of Juvenile Leaves and Adult Leaves Regulated

Juvenile leaves and adult leaves differ in abaxial trichome production, length/width ratio, and degrees of serration on the leaf margin [5,10,31]. Juvenile leaves do not produce abaxial trichomes whereas adult leaves do [5]. Trichomes are initiated by a MYB-bHLH-WD40 protein complex, which includes the R2R3-MYB protein GLABRA1 (GL1). The temporal and spatial distribution of trichomes on juvenile leaves and adult leaves is in part determined by the temporal and spatial regulation of *GL1*. *miR172b* is a direct target of SPL9, and miR172 directly represses members of the *APETALA2-like* (*AP2*-like) gene family, *AP2*, *TARGET OF EAT1* (*TOE1*), *TOE2*, *TOE3*, *SCHLAFMUTZE* (*SMZ*), and *SCHNARCHZAPFEN* (*SNZ*) [10]. TOE1 binds to the 3′ region of *GL1* [44,45], and KANADI1 (KAN1), an abaxial polarity gene [46], also binds to the 3′ end of *GL1*. Further protein–protein analysis showed that KAN1 physically interacts with TOE1 [44]. A chromosome conformation capture (3C) assay showed that the KAN1-TOE1 repressor complex was bound to the 5′ region of *GL1* [44]. Thus, in juvenile leaves when levels of miR156 are high, and levels of TOE1 are also high because the direct target of miR156-SPL module, *miR172b*, is low, the KAN1-TOE1 complex represses the expression *GL1* on the abaxial side of leaves. In adult leaves when miR156 levels have dropped, *TOE1*, at the end of the miR156-SPL-miR172-TOE1 module, is downregulated, the repressor complex is no longer present and *GL1* is derepressed (Figure 2a). 

In *Arabidopsis*, leaves 1 and 2 have smooth margins; subsequent leaves become more and more serrated as the plant ages. Leaf margin serration is largely controlled by CUP-SHAPED COTYLEDON (CUC) and TEOSINTE BRANCHED1/CYCLOIDEA/PCF (TCP) transcription factors. CUC2 but not CUC3 is a target of miR164, but both are expressed at the boundaries of incipient serrations [47,48]. miR164a mutants or plants expressing an miR164-resistant *CUC2* produce leaves with serrated margins while mutations in *CUC2* or overexpression of miR164 suppress serrated leaf margins [47,48]. A gain of function mutation in miR319 (JAW-D), which targets TCP transcription factors, also produces leaves with serrated margins. TCP4 interferes with the dimerization of CUC proteins [49], suggesting that the serrated leaves in JAW-D might result from decreased TCP activity and increased CUC dimerization. SPL9 physically interacts with TCP and competes with CUC proteins in binding to TCP. Therefore, the increased serration at the leaf margin as plants age may be explained by physical interactions among SPL9-TCP-CUC proteins. In early-arising leaves where the activities of miR156-targeted SPLs are low, TCP proteins bind to CUC proteins and prevent the dimerization of CUC proteins, which induces smooth margins. As the plant ages, miR156 levels decline and expression of miR156-targeted SPLs increase. SPLs bind to TCP, releasing CUC proteins from the inhibition by TCPs and inducing serrated leaf margins (Figure 2a). However, the interactions between SPL and TCP proteins does not seem to be the only pathway for margin serration. TCP3 binds to a variety of genes in regulating leaf margin serration and shoot apical meristem activity [50], and TCP may regulate leaf margin serration by multiple pathways. 

The time interval between successive leaf initiation events (plastochron) is shorter during the juvenile phase than it is during the adult phase [51]. Plants with a prolonged juvenile phase, such as those overexpressing miR156 and those with mutations in multiple SPLs have a faster leaf initiation rate [10,11]. The increased leaf initiation rate in *35S::MIR156F* mimics that of *cyp78a5*, the rice plastochron ortholog in *Arabidopsis*. However, molecular and genetic analysis of *CYP78A5* and *SPL*s indicated that they do not appear to be downstream targets of each other. Rather, both regulate the cell division marker Histone H4 to increase the rate of cell division within the shoot apical meristem (Figure 2a) [51].

Solid lines represent direct interaction, dashed lines indicate indirect interaction or it is not known if the action is direct or indirect. Arrows represent activation, T-bars represent repression. Activation of miR156 results in late vegetative phase change, and activation of SPLs results in accelerated vegetative phase change. Repression of miR156 results in accelerated vegetative phase change, and repression of SPLs results in late vegetative phase change. 

## 4. What Controls the Timing of the Vegetative Phase Change

### 4.1. Signals from the Embryo Promote Vegetative Phase Change

Maintaining high levels of miR156 in the embryo may be essential for embryo development, as *SPL10* and *SPL11* were strongly upregulated in the lethal *dcl1* embryo [52]. FUSCA3 (FUS3) plays important roles in establishing and maintaining embryonic identity during seed development [53,54], and *fus3* mutants not only have defects in embryo development but also exhibit early production of abaxial trichomes [55]. Chromatin immunoprecipitation (ChIP) analysis of *FUS3::FUS3-c-myc fuc3-3* transgenic plants showed that FUS3 binds to *MIR156A* and *MIR156C* directly, and transcripts from these two loci, *pri-miR156a* and *pri-miR156c* were significantly downregulated in *fus3* developing seeds [56]. Mutations in *CENTER CITY (CCT)/ MEDIATOR12 (MED12)* and *GRAND CENTRAL (GCT)/MED13*, two mediator cyclin-dependent kinase module genes that promote specification of central and peripheral identity and the globular-to-heart transition during embryogenesis [57], delay the timing of the vegetative phase change. miR156 levels were significantly higher in *gct* and *cct* mutants, and these mutants exhibit a delay in the production of abaxial trichomes [58]. The CCT/GCT pathway appears to act independently of the sugar pathway [59]. 

Abscisic acid (ABA), a plant hormone that inhibits seed germination, induces the accumulation of miR159 during seed germination [60]. miR159 is expressed in mature embryos, and mutations in *mir159a/b* result in small and irregular seeds and upward curling leaves [61]. The *mir159a/b* double mutant was reported to produce increased numbers of small and round leaves without abaxial trichomes, indicating a delay of the vegetative phase change [62]. Consistent with this, *pri-miR156a/pri-miR156c* are derepressed in the *mir159a/b* double mutant. miR159 represses the R2R3 MYB domain transcription factors *MYB33* and *MYB65* directly [61], and miR159-resistant *MYB33* (*mMYB33*) mimics the *mir159a/b* double mutant in delaying abaxial trichome production and derepressing *pri-miR156a/pri-miR156c* [62]. Furthermore, ChIP analysis showed that MYB33 binds to *MIR156A/MIR156C* and *SPL9* directly [62]. Together, the regulation of vegetative phase change by these embryo regulators suggest that there may be a network coordinating embryo and vegetative development and that signals from the embryo regulators may continue to act during vegetative development (Figure 2b,c).

### 4.2. Sugar from Leaves is a Signal for Vegetative Phase Change

miR156 levels are high in the juvenile phase and dramatically reduced in the adult phase [10,63]. It is very important to understand the molecular mechanisms that lead to miR156 downregulation. Studies using maize shoot apices with leaf primordia of different sizes suggest that the signal for vegetative phase change originates from leaves. In maize, leaf 6 and leaf 7 are juvenile-to-adult transition leaves and leaf 8 is normally an adult leaf [64]. Culturing shoot apices when leaf primordia 8 is 1.7–3.2 mm long resulted in leaf 8 and several subsequent leaves having juvenile traits at the base of the leaf blade. Additionally it was shown in *Arabidopsis* and tobacco plants that defoliation attenuates downregulation of miR156 and delays vegetative phase change [65]. A product of photosynthesis may be the signal that induces downregulation of miR156, as miR156 is upregulated in a chlorophyll-deficient mutant *chlorina1-4* and the vegetative phase change is delayed in this mutant [66]. Sugar is the major output of photosynthesis, and plants treated with sucrose, glucose, or fructose repress *pri-miR156a/pri-miR156c* transcription [66,67]. Trehalose 6-phosphate (T6P) is a sugar status sensor that functions as a signaling molecule in plant cells. Knocking down expression of *TREHALOSE PHOSPHATE SYNTHASE1* (*TPS1*), which encodes an enzyme catalyzing synthesis of T6P, attenuates downregulation of *pri-miR156a/pri-miR156c* and delays vegetative phase change [67]. Activation of *TPS1* in an inducible line triggers downregulation of miR156. Interestingly, levels of sucrose, but not glucose nor fructose, are increased in transgenic plants with reduced activites of TPS1 [68]. The above results indicate that sugar promotes the adult phase by downregulating miR156. 

To investigate how sugar mediates vegetative phase change in trees and crops, Lawrence et al. (2020) measured the photosynthetic rates in juvenile leaves and adult leaves of *Populus*, maize and *Arabidopsis* in different conditions [69]. They found that *Populus* and *Arabidopsis* adult leaves had higher photosynthetic rates per leaf area than their juvenile leaves and juvenilized leaves overexpressing miR156. However, photosynthetic rates per mass, especially in low light conditions, are higher in juvenile leaves than in adult leaves in all three species [69]. This work in three phylogenetically diverse species showed that in full sunlight conditions, higher photosynthesis rate is correlated with adult leaves, confirming the important role of sugar in vegetative phase change. In lower canopy conditions, juvenile leaves require a high photosynthetic rate for compensation and survival. Together, these studies suggest that sugar produced in leaves induces the downregulation of miR156 to mediate vegetative phase change, and that the sugar status sensor T6P is involved in mediating the effect of sugar on vegetative phase change (Figure 2c). 

### 4.3. Signals from the Shoot Apical Meristem Prevent Precocious Vegetative Phase Change

Plants overexpressing *miR156a* (*35S::MIR156A*) have a wider shoot apical meristem (SAM), while plants overexpressing the miR156 target mimic (*35S::MIM156*) have a narrower SAM, suggesting that factors modulating SAM size might regulate vegetative phase change [50]. This is supported by the observation that mutants with defects in meristem maintenance, such as *wuschel* (*wus*), *shootmeristemless* (*stm*), *paused* (*psd*), and the triple mutant *Arabidopsis thaliana homeobox1 pennywise pound-foolish* (*ath1 pny pnf*) all produce adult leaves earlier than WT [70] (Figure 2a). Transcripts of *pri-miR156a/pri-miR156c* were down-regulated in *wus* and *psd*, and transcripts and proteins of *SPL9* were upregulated [70]. Expression of *WUS* is upregulated in *spl2/9/10/11/13/15* mutant, suggesting that SPLs repress *WUS* in the SAM to modulate SAM size. miR156 and its targeted-SPLs are expressed in the SAM and leaf primordia [10,11,43,50,62], and it is not clear where their expression determines leaf identity. Levels of miR156 and miR157 were dramatically reduced in the *mir156a/c mir157a/c* quadruple mutant, and vegetative phase change is accelerated to leaf 3 in this quadruple mutant while it is leaf 5 in WT [36]. Expressing *MIR156A* under the control of the SAM promoters *WUS* and *STM* in the *mir156a/c mir157a/c* quadruple mutant slightly restored vegetative phase change timing, while expressing *MIR156A* under the control of the promoter of the primordia gene *AINTEGUMENTA* (*ANT*) or the promoter of the SAM gene *FD* in the quadruple mutant phenocopied the *35S::MIR156A* plant. Similarly, expressing the miR156 target mimic (*MIM156*), that blocks the activities of miR156, using the *WUS*, *STM* and *FD* promoters slightly accelerated the vegetative phase change, while expressing the miR156 target mimic using the *ANT* promoter resembled *35S::MIM156* transgenic plants [70], suggesting that the activities of miR156 in leaf primordia might be functionally more important. Since *WUS* is expressed in a small number of cells, called the organizing center, within the SAM [71], how does the expression of *MIR156A* in the *WUS* domain rescue leaf identity in *mir156a/c mir157a/c* quadruple mutants? Analysis of miR156 sensitive *SPL9::GUS* in the *mir156a/c mir157a*/c quadruple mutant and *WUS::MIR156A mir156a/c mir157a/c* plants showed that SPL9 in leaf primordia is repressed by *WUS::MIR156A*, suggesting that miR156 functions non-cell autonomously to repress *SPL9* [70]. WUS forms a feedback loop with CLAVATA1 (CLV1) and CLV3 in the SAM to maintain proper size of the SAM [71]. The SAM of *clv1* and *clv3* mutants are bigger than that of the WT [71], and production of the first leaf with abaxial trichomes is delayed in *clv1* and *clv3* mutants. However, the delayed abaxial trichome production in *clv1* and *clv3* mutants results from a faster leaf initiation rate, rather than from changes in SPL expression [37]. Together, the results from Fouracre et al. suggest that miR156 acts non-cell autonomously in mediating the juvenile-to-adult phase change. 

### 4.4. Hormones Act on Vegetative Phase Change 

Hormones have essential roles in promoting phase transitions, such as seed germination, seedling morphogenesis and floral induction, which suggests that they may contribute to vegetative phase change [72,73,74]. The best-known hormone that controls vegetative phase change is gibberellic acid (GA). Vegetative phase change defects in the GA-deficient maize mutant *dwarf* are rescued by treating the plants with GA [75]. GA is perceived by GIBBERELLIN INSENSITIVE DWARF1 (GID1) and regulates target gene expression through degradation of DELLA repressor proteins [76,77]. The DELLA protein REPRESSOR OF GA1-3 (RGA) physically interacts with SPL3 and SPL9 and inhibits the activities of SPL3 and SPL9. Treating plants with GA results in RGA degradation and activation of SPLs [78]. Abscisic acid (ABA) affects vegetative phase change by upregulating two miR156 primary transcripts *pri-miR156a/pri-miR156c* [60,62]. The ABSCISIC ACID INSENSITIVE3 (ABI3) transcription factor binds directly to *MIR156A*, *MIR156C* and *MIR156D* but not *SPL10* nor *SPL11* [79]. Other hormones that may affect vegetative phase change are the defense hormone jasmonic acid (JA) and auxin [80]. A role for JA in vegetative phase change is suggested by the direct interaction of SPL9 with JA ZIM domain (JAZ) proteins that are involved in herbivore defense [12]. Mutations in two auxin responsive factors *ARF3* and *ARF4*, suppress the precocious vegetative phase change in *zip* (*ago7*), and the *arf3 arf4* double mutant is delayed in vegetative phase change [29]. These studies suggest that GA, ABA, JA and auxin contribute to vegetative phase change timing, but how these hormones are integrated to regulate vegetative phase change timing is largely unknown (Figure 2c). 

### 4.5. Endogenous Epigenetic Factors Regulate Vegetative Phase Change 

The formation of nucleosomes (146-147 bp of DNA wrapped around a histone octamer) is the first level in compaction of the length of DNA that can prevent access of RNA polymerase to the promoter, thereby preventing transcription [81]. Chemical modifications on the tails of the core histones can either result in further condensation of the chromatin and gene silencing (by repressive marks) or relaxation of the chromatin and transcription (by activating marks) [81]. Eight loci, *MIR156A-MIR156H*, encode miR156 in Arabidopsis. Genetic analyses indicates that *MIR156A/MIR156C* are the major sources for mature miR156 [36,66]. The *MIR156A/MIR156C* loci are temporally marked by H3K27ac, H3K27me3, and K3K4me3, suggesting that *MIR156A/MIR156C* are subject to epigenetic regulation [11,82,83]. The abundance of the active markers H3K27ac and H3K4me3 decreases temporally, while the abundance of the repressive marker H3K27me3 increases temporally, accompanied with temporal downregulation of *pri-miR156a/pri-miR156c* [63,82]. Mutations in *PICKLE* (*PKL*), a CHD3 chromatin remodeler, delay vegetative phase change by decreasing nucleosome occupancy, increasing the abundance of H3K27ac and reducing the abundance of H3K27me3 at *MIR156A/MIR156C* [62]. Histone 3 lysine-27 trimethylation is mediated mainly by Polycomb Repressive Complex1 (PRC1) and PRC2. The methyltransferase SWINGER (SWN) in PRC2 functions synergistically with PKL to modulate the abundance of H3K27me3 and the transcripts of *pri-miR156a/pri-miR156c* [63]. Mutations in the PRC1 component *AtBMI1a/b* caused a reduction in the abundance of H2A ubiquitination and H3K27me3 at *MIR156A/MIR156C*, derepressing *pri-miR156a/pri-miR156c* and delaying vegetative phase change [84]. Another chromatin remodeler, the SWI2/SNF2 ATPase BRAHMA (BRM), however, acts antagonistically to SWN in vegetative phase change by decreasing nucleosome occupancy near the transcription start site of *MIR156A* [83]. BRM interacts with two BRAHMA-interacting proteins1 (BRIP1) and BRIP2 to modulate chromatin structure [85], and it is likely that this BRM–BRIP complex regulates chromatin structure at *MIR156A/MIR156C*. *MIR156A/MIR156C* chromatin is also marked by H3K4me3, which is an active marker for many loci. A SET domain protein ATXR7 that has H3K4 methyltransferase activity binds to *MIR156A* directly to modulate the abundance of H3K4me3 [82]. The abundance of H3K4me3 is also modulated by the SWR1 complex, whose activity mediates the histone exchange of H2A.Z for H2A [86]. Mutations in the SWR1 complex components *ACTIN-RELATED PROTEIN6* (*ARP6*) or *SERRARED LEAVES AND EARLY FLOWERING* (*SEF*) resulted in reduced H2A.Z levels and reduced H3K4me3 levels at *MIR156A/MIR156C*. These reductions, however, only caused an increase in H3K27me3 at *MIR156A*, not at *MIR156C*, suggesting that the abundance of the repressive marker H3K27me3 is not necessarily associated with the abundance of the active marker H3K4me3 [82] (Figure 3b, Table 1). 

The miR156-targeted SPLs are also subjected to epigenetic regulation. Two PRC1 components AtRING1A/AtRING1B were reported to suppress precocious vegetative phase change by repressing miR156-targeted SPLs [87]. miR156 levels did not change in the *atring1a atring1b* double mutant although levels of miR156-targeted SPLs were increased in the double mutant. The upregulation of miR156-targeted SPLs could be attributed to a reduction in the levels of H2A monoubiquitin at these SPL loci [87]. Mutation in the SAGA-like Histone Acetyltransferase of the GNAT/MYST superfamily 1 (HAG1) delayed vegetative phase change by downregulating *SPL3* and *SPL9*, whereas levels of miR156 were not changed in *hag1*. HAG1 binds to SPL9 directly and mediates the abundance of H3Ac at *SPL9* [88]. Epigenetic regulation of miR156-targted SPLs might be responsible for the partial inability to restore juvenile status (rejuvenation) by inducing miR156 in adult plants [89]. Together, these studies showed that several epigenetic markers are present on miR156 and SPLs. Further studies need to be carried out to reveal what mediates the temporal enrichment of these epigenetic markers and how these different epigenetic markers coordinate to regulate vegetative phase change (Figure 3d, Table 1). 

## 5. Is Vegetative Phase Change a Prerequisite for Floral Induction

Vegetative phase change precedes floral induction and an important question is whether vegetative phase change is required for floral induction [90]. In *Arabidopsis*, SPL3, SPL9 and potentially other miR156-targeted SPLs directly activate the expression of the floral induction genes *SUPPRESSOR OF OVEREXPRESSION OF CONSTANS 1* (*SOC1)*, *FRUITFUL* (*FUL*), and the floral meristem identity genes *LEAFY* (*LFY*) and *APETALA1* (*AP1*) to promote flowering [43,91]. The cauline leaves of *ful* are rounder than those of wild type and resemble those of the *spl2/10/11* triple mutant [92]. Molecular analysis showed that *FUL* is upregulated in plants overexpressing *SPL10*, indicating that FUL may function downstream of SPLs in mediating cauline leaf development [92]. The bZIP transcription factor FD interacts with both the flowering activator FLOWERING LOCUS T (FT) and the flowering repressor TERMINAL FLOWER1 (TFL1); these two complexes act antagonistically on a common set of targets, such as *LFY*, *AP1*, and other flowering genes to induce flowering [93,94,95]. SPL3/SPL4/SPL5 interact with the floral activator FD to stimulate flowering under normal light conditions, and FAR-RED ELONGATED HYPOCOTYL3 and FAR-RED IMPAIRED RESPONSE1 prevent SPL3/SPL4/SPL5 activity in shady conditions to suppress flowering [96,97]. *Arabis alpine*, a perennial relative of annual *Arabidopsis thaliana*, requires an extended vegetative growth before induction of flowering by vernalization [98]. Activation of *AaSQUAMOSA PROMOTER BINDING PROTEIN LIKE15* (*AaSPL15*), a master regulator for the adult phase, is a prerequisite for floral induction [99,100]. However, earlier work in maize gave a different result. *Tp2*, which has a prolonged juvenile phase, responds to floral induction similar to WT, indicating that the delay in the vegetative phase transition does not delay reproductive maturation [90]. Together, these studies suggest that the juvenile-to-adult vegetative phase change is not necessarily a prerequisite for floral induction in all flowering plants.

## 6. Conclusions

Vegetative phase change, essential for proper development of plants, is driven by the evolutionarily conserved miR156-SPL module. Since this module also controls root development, insect resistance, plant immune responses, stress responses, and crop yield [11,12,13,14,15,16,17], a deeper understanding of this process and how it is regulated will facilitate efficient improvements of agricultural traits. This review summarizes the most recent advancements that regulate this developmental transition.

Temporal regulation of miR156 levels drives the transition from the juvenile phase to the adult phase. This regulation correlates well with the temporal binding of the PRC2 complex on *MIR156A/MIR156C*, but the mechanisms responsible for changes in PRC2 complex occupancy remain unknown. GA contributes to vegetative phase change through the release of SPLs from their inhibitory interaction with the DELLA proteins. While there is evidence that other hormones including auxin, ABA, and jasmonic acid are also involved in vegetative phase change, how these hormones coordinate vegetative phase change remains unclear. Plants adjust their developmental program in response to stress conditions or environmental changes to maximize their survival and fitness. Several environmental conditions are associated with regulation of miR156 expression. For example, it has been reported that production of miR156 is correlated with stress treatments, such as heat treatment and cold treatment [101,102]. In addition, one of the PHYTOCHROME-INTERACTING FACTORS (PIFs), PIF5, which is induced in low red/far red and low blue light [103], binds directly to the genomic loci of some miR156 precursors to mediate shade avoidance (a condition of low red/far red ratio) [104]. It is not known how vegetative phase change is mediated in different light conditions. Understanding how plants coordinate the juvenile-to-adult phase transition in response to light, temperature, and nutrient availability will be important for understanding how plants balance development and environmental responses.

## Figures and Tables

**Figure 1 ijms-21-09753-f001:**
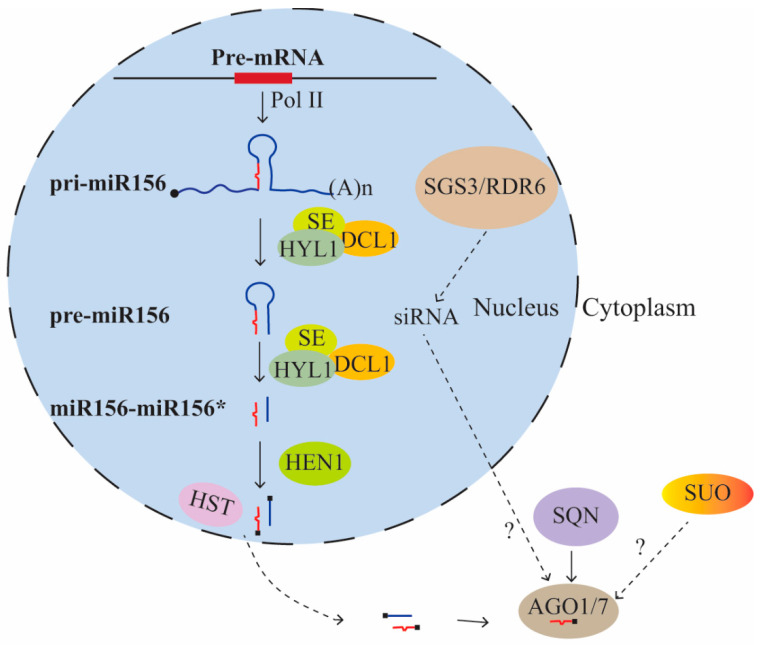
Biogenesis of miR156.

**Figure 2 ijms-21-09753-f002:**
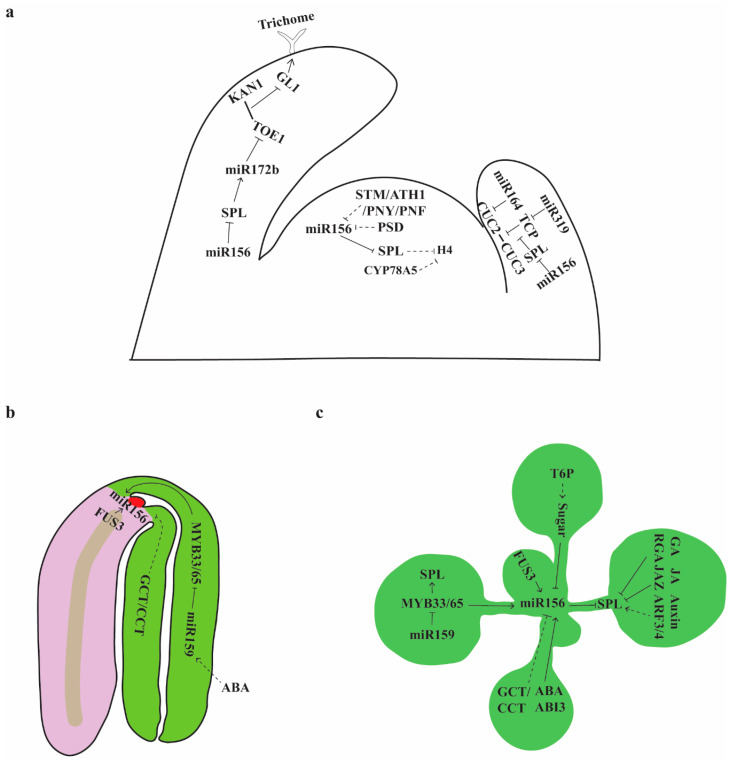
Spatial and temporal regulation of vegetative phase change.(**a**) Shoot apical meristem (SAM) regulators and vegetative phase change. miR156 and *SPL*s are expressed in the shoot apical meristem, leaf primordia, expanding leaves, and expanded leaves. In juvenile tissues where levels of miR156 are high, *SPL*s are repressed. In adult tissues where levels of miR156 are low, *SPL*s are derepressed. WUS, STM, ATH1, PNY, PNF and PSD repress miR156 in the SAM. In leaves, miR156-targeted SPLs activate *miR172b*, which repress a group of AP2 transcription factors. TOE1, a member of the miR172-targted AP2 family, physically interacts with leaf abaxial identity regulator KAN1 and forms a repressing loop at the trichome initiation gene *GL1*. *CUC* genes are expressed at the boundary of leaf primordia and the SAM, and physically interact with each other to promote serration in expanded leaves. *CUC2* is repressed by miR164, miR319-targeted TCP interferes with the interaction between CUC2 and CUC3, and miR156-targted SPLs bind to TCP to release CUC from TCP to promote leaf serration. SPL and CYP78A5 repress H4 to control leaf initiation.(**b**) Regulation of miR156 in embryo. miR156 is highly expressed in the embryo by the activities of MYB33/65 and FUS3, while GCT/CCT repress miR156. (**c**) Regulation of miR156 and SPL in seedling. MYB33/65, FUS3, and GCT/CCT continue to act in seedlings and sugar produced from photosynthesis represses miR156. Hormones are involved in vegetative phase change by acting on SPLs directly or indirectly. The gibberellic acid (GA) signaling molecule RGA and the jasmonic acid (JA) signaling molecule JAZ directly interact with miR156-targeted SPLs and repress their activities. The abscisic acid (ABA) signaling molecule ABI3 represses SPL via activating miR156. The auxin signaling molecules ARF3 and ARF4 suppress precocious vegetative phase change and it is not known if they act on SPLs directly or indirectly.

**Figure 3 ijms-21-09753-f003:**
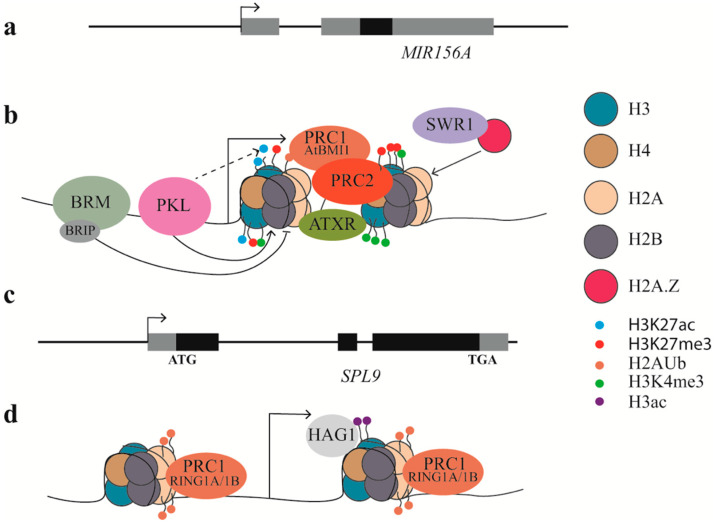
Epigenetic regulation of vegetative phase change. (**a**,**c**) Schematic diagrams of *MIR156A* and *SPL9*. Solid black boxes represent stem-loop structure of *MIR156A* or exons of *SPL9*, solid grey boxes represent transcripts that were not functional or untranslated. Arrow heads indicate transcription start site.(**b**) Epigenetic regulation of *MIR156A*. The formation of nucelsomes after the transcription start site prevents transcription. Chemical modifications on the tails of the core histones can either result in further condensation of the chromatin and gene silencing or relaxation of the chromatin and transcription. PKL facilitates the formation of a nucleosome after the transcription start site, while BRM prevents the formation of a nucleosome. PKL potentially interacts with other factors to remove acetylation at H3K27, allowing methylation at H3K27. PRC2 facilitates H3K27me3 directly while PRC1 (AtBMI1) facilitates ubiquitination (Ub) of H2A and H3K27me3. SWR1 complex facilitates the exchange of H2A to H2A.Z, which favors methylation at H3K4. ATXR proteins directly deposit H3K4me3. (**d**) Epigenetic regulation of *SPL9*. PRC1 (RING1A/1B) facilitates Ub of H2A at SPL9, while HAG1 facilitates acetylation of H3 at *SPL9*.

**Table 1 ijms-21-09753-t001:** Epigenetic factors regulating vegetative phase change (VPC).

Epigenetic Factors	Family or Complex It Belongs to	Function	Target	Promote or Suppress VPC	References
PKL	CHD3 protein	promote nucleosome/H3K27me3	miR156	promote	[63]
BRM	SWI2/SNF2 protein	repress nucleosome/H3K27me3	miR156	suppress	[82]
SWN	PRC2 complex	promote H3K27me3	miR156	promote	[63]
ARP6/SEF	SWR1 complex	promote H2A.Z and H3K4me3	miR156	suppress	[82]
ATXR7	SET domain protein	promote H3K4me3	miR156	suppress	[82]
AtRING1A/B	PRC1 complex	promote H2Aub	SPL	suppress	[87]
AtBMI1A/B/C	PRC1 complex	promote H2Aub and H3K27me3	miR156	promote	[84]
HAG1	SAGA-like complex	Promote H3Ac	SPL	promote	[88]

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
