# Peer review of "Juvenile Leaves or Adult Leaves: Determinants for Vegetative Phase Change in Flowering Plants"

_ijms, 2020, doi:10.3390/ijms21249753_

Round 1
Reviewer 1 Report
The manuscript “Juvenile leaves or adult leaves: Determinants for vegetative phase change in higher plants” by Manuela and Xu provides a rather complete and understandable review of the current knowledge on the transition from juvenile to adult leaves. There are no major remarks to be made.
However there are some things that could improve the manuscript. The authors could make it more accessible to the non-specialist, and some suggestions for this have been made in the attached document.
The figure 1 is hardly readable in print, changing the purple background to something less hard will make this easier.
Besides that there are only comments of the editorial type, the correct use of italics for species names and the use of italics, capitals, … to differentiate genes, transcripts, proteins and mutants from each other. On the last point there are some flaws that cause confusion.
Finally some of the used abbreviations would better be defined, one can mention SAM and pri-miR but there are others that may not be obvious for people not directly working on this topic or using similar approaches to study biology.

Author Response
Dear Reviewer,
Thank you very much for the detailed comments on the manuscript. These comments are very valuable for us to improve our manuscript. We have made changes to the manuscript as much as we could, and below is a summary of our changes:
1, Color of Figure1. We have changed the color for nucleus, cytoplasm, SQN and exon beside miR156 such that it is better for reading. Besides this, we have increased font in figures so that they are more readable, the sign beside AGO1/7 is too small to read, and we replaced it with black boxes and explained in figure legend that it represents methylation of the 3’end. And we have removed the title on top of each figure.
2, We have changed the form of many names, such as “Arabidopsis thaliana” and “Arabidopsis” to italic form “Arabidopsis thaliana” and “Arabidopsis”. And we have changed the words suggested by this reviewer.
3, Definition for some of the abbreviations are missing, such as SGS3, RDR6, TFL1, FT, DCL1, SE, HYL1, HEN1, RGA and HAG1. We added the full names for those.
4, line 128, miR172. miR172 is the product of miR172b. Here it refers to miR172, not miR172b.
5, We have updated Figure 2 and figure legend for Figure 2.
6, line 234-line242 are rephrased. It is correct that the connection between sugar and photosynthesis is not clear in the old description. We rephrased the sentences so that the connection is clear. Please see updated manuscript for details.
Sincerely
Mingli
Reviewer 2 Report
Manuscript by Manuela and Xu entitled “Juvenile leaves or adult leaves: determinants for vegetative phase change in higher plants” intended for publication in International Journal of Molecular Sciences is an interesting paper, however I think that in the present form is not ready to publish and needs improvements.
I have mainly minor remarks. The Authors should improve Keywords – remove title repetitions. The Authors could specify/improve purpose of the study. The Authors should improve Figures visibility/presentation – remove titles e.g. “Figure 1”, use larger fonts (especially at the Figs. 1-2), some captions could be described in a more detailed way, e.g. explain used abbreviations. In addition, there are a lot of different small mistakes in the text of manuscript, especially in Reference list (e.g. lines 187, 273, 365, 366, 424, 425, 443, 444, 447, 474, 508, 525, 535, 548, 550, 561, 563, 566, 582, 588, 595, 599, 612, 622, 629, 633, 636), that need to be corrected by Authors.
Author Response
Dear Reviewer,
Thank you very much for your valuable comments on the manuscript. We have made changes to the manuscript as much as we could, and below is a summary of our changes:
1. We have increased font of in figures so that they are more readable.
2. We have changed the key words to be: miR156, SPL, abaxial trichome, juvenile phase, adult phase. These are the topics that we have been discussing throughout the manuscript.
3. We have added full names for the abbreviations, such as SGS3, RDR6, TFL1, FT, DCL1, SE, HYL1, HEN1, RGA and HAG1.
4. The purpose of the study is enriched at the Abstract and Introduction sections. Please see the updated manuscript for details.
5. We have updated all the lines indicated below: 187, 273, 365, 366, 424, 425, 443, 444, 447, 474, 508, 525, 535, 548, 550, 561, 563, 566, 582, 588, 595, 599, 612, 622, 629, 633, 636. Please see the updated manuscript for details.
Thank you again for reviewing this manuscript!
Sincerely
Mingli
Reviewer 3 Report
Dear Authors,
the manuscript entitled “Juvenile leaves or adult leaves: determinants for vegetative phase change in higher plants” is well written despite the complexity of the theme, so it deserves to be accepted with only the suggestion of making some really small changes.
The authors analyse the subject of the review very well. Given the great link between miR156 and miR172, I would have explored this part further.
Abbreviations: actually, it is limited to the most known and obvious abbreviations, please add all other abbreviations used in the text.
Best regards
Author Response
Dear Reviewer,
Thank you very much for reviewing this manuscript. Your comments are a great encouragement and we will keep up. We went through the manuscript carefully and have added full names for the abbreviations, such as SGS3, RDR6, TFL1, FT, DCL1, SE, HYL1, HEN1, RGA and HAG1.
We have rephrased the manuscript minorly, please see the updated manuscript for details.
Thank you again!
Sincerely
Mingli